# Is it all in your head? Reducing virtual reality induced cybersickness by pleasant odor imagery

Luca Fantin[1,2,3], Gabriela Hossu[1,4], Cécile Rumeau[5,6], Guillaume Drouot[4], Hadrien Ceyte[2]*

1 Université de Lorraine, Inserm, IADI, Nancy, France, 2 Aix Marseille Université, CNRS, ISM, Marseille, France, 3 Normandie Univ, UNICAEN, ENSICAEN, CNRS, GREYC, Caen, France, 4 CHRU-Nancy, Inserm, Université de Lorraine, CIC, Innovation Technologique, Nancy, France, 5 CHRU-Nancy, Université de Lorraine, Service ORL, Nancy, France, 6 Université de Lorraine, DevAH, Nancy, France

* hadrien.ceyte@univ-amu.fr

## Abstract

Although immersive technologies such as virtual reality are constantly growing for personal and professional purposes, their use can often induce a transient state of discomfort known as cybersickness, resulting in numerous symptoms and perceptive-motor vulnerability. In an attempt to develop leads to mitigate cybersickness, encouraging findings have reported decreased symptoms during the presentation of pleasant smells. However, the diffusion of smells in ecological settings is very challenging. An interesting alternative could reside in odor imagery (OI), known for its neurophysiological, behavioral and psychological similarities with odor perception. The aim of this study was therefore to determine the effects of pleasant OI on virtual reality induced cybersickness. Thirty participants performed two 14-minute virtual reality sessions simulating a first-person view from a boat. During the second session we added a picture at the center of the visual field, allowing for pleasant and intense OI based on individualized psychometrical measures. Participants were instructed to focus on the smell evoked by this picture. For both immersions, cybersickness was characterized by the evolution of scores on the Simulator Sickness Questionnaire, and duration of immersion. Our results show that both measures were positively affected by pleasant OI, indicating a decreased intensity of cybersickness symptoms associated with a longer tolerance of the virtual environment. We suggest the observed effects could be mediated by emotional regulation mechanisms driven by pleasant OI, alleviating cybersickness in a similar way to pleasant odor perception. These findings could open the door to new applications of pleasant sensory imagery as strategies to alleviate transient states of discomfort in immersive technologies or perhaps motion-induced sickness.

## Introduction

The use of immersive virtual technologies, and particularly virtual reality (VR) has astoundingly grown over the past decade. While VR is mostly used for leisure, it is becoming

**Data availability statement:** All data used for the descriptive statistics or formal analyses are publicly available at https://doi.org/10.6084/m9.figshare.26927116.v1

**Funding:** This work is co-funded by the French State-Region contract CPER 2015-2020 (Contrat de Plan Etat Région– IT2MP Innovations Technologiques, Modélisation et Médecine Personnalisée), by the European Union through the European Regional Development Fund "FEDER-FSE Lorraine et Massif des Vosges 2014-2020" and by the CHRU of Nancy, France. The funders had no role in study design, data collection and analysis, decision to publish, or preparation of the manuscript.

**Competing interests:** The authors have declared that no competing interests exist.

essential in the fields of industry [1], or for pedagogical purposes in surgeons [2] or pilots [3]. In the field of research, VR can also be of great interest since it offers controlled environmental conditions, and can negate visual distractors. It is therefore used to study cognitive-motor behaviors in controlled visual environments with limited logistical constraints. However, the use of VR can quite often lead to a state of discomfort known as cybersickness. Usually defined as the result of a cognitive-sensory conflict [4], cybersickness can manifest through several symptoms categorized in nausea, oculomotor symptoms, and disorientation [5,6]. Furthermore, recent findings have shown that cybersickness induces an alteration of the importance attributed to visual cues in geocentric perception, which could induce posture-locomotor vulnerability in some at-risk populations [7]. Although numbers vary, it has been reported that in some cases up to 80% can be subject to this state of discomfort in VR [8–10]. Given the place of VR nowadays and the consistently increasing number of users, it has become a public health issue to determine the factors capable of reducing the risk of cybersickness. In this regard, the occurrence of cybersickness can admittedly be modulated by multiple factors related to hardware, the cognitive load of the task performed in VR [11], the virtual environment itself or characteristics related to the user. To this day however, most work has focused on the latter (i.e., human factors). For example, higher intensity of cybersickness has been shown in women [12] and individuals with higher static field dependence [7] or sensitivity to visual cues broadly speaking [13]. On the other hand, greater experience with video games has been associated with decreased cybersickness [14]. Overall, these human factors are difficult or impossible to control, although they may help with setting recommendations. An alternative which may prove more efficient in preventing cybersickness on short notice resides in environmental factors. To this day only little is known about how controlling the environment can help alleviate cybersickness, and more generally speaking all forms of visually-induced states of discomfort. In one notable study [15], it was shown that the diffusion of a pleasant odor (smell of rose) during a first-person cycling simulation, significantly reduced the intensity of symptoms. The interpretation of these results is that the pleasant odor elicited a positive emotional state in participants, reorienting their attention towards these emotions rather than the nauseogenic attribute of the visual stimulation [16]. Although this finding was a significant advance, the diffusion of smells in ecological situations can be limited by several factors, such as the cost and technicity of the material used for odor diffusion, or psychophysical issues related to subjectivity of odor pleasantness and intensity [17].

In this study we propose to study another form of olfaction, namely odor imagery (OI), which can be described as the mental simulation of a smell in absence of the corresponding physical stimulation [18,19]. Though its existence has been questioned in the past, the ability to generate odor images is now commonly accepted and can be facilitated by the presence of complementary sensory information such as visual cues, especially those providing more semantical context (i.e., pictures and words) [20]. Across the years, numerous neurophysiological, psychological and behavioral similarities between odor perception and OI have been demonstrated [21–24]. In particular, odor perception and OI seem to share neural pathways, both involving brain regions with a role in emotional regulation [25–27]. Moreover, activity in brain regions such as the insula which has strong limbic connections, can be modulated by hedonicity of both perceived smells and mental odor images [28]. Therefore, if cybersickness can be alleviated by emotional states mediated by olfaction, performing pleasant OI during VR immersion may provide similar effects while being free of the technical and material limits of odor diffusion. The aim of the present study was thus to determine the impact of pleasant OI on VR-induced cybersickness, expressed by the intensity of symptoms and the duration of VR immersion.

## Materials and methods

This study received ethical approval by the local ethics committee ("CANOE" clinical trial; ClinicalTrials.gov Identifier: NCT05308433). The recruitment period began on April 14th, 2022 and ended on July 12th, 2022. Thirty healthy volunteers (14 women, 16 men; mean age = 22.5 years old; SD = 3.4) took part in this study. Oral and written informed consents were obtained from each participant before inclusion. Non-inclusion criteria related to this part of the clinical trial were any olfactory disorder or non-corrected visual impairment. All participants were characterized as being sufficiently able to imagine smells by obtaining a score strictly below 2.5/5 on the *French Vividness of Olfactory Imagery Questionnaire* (fVOIQ) [29]. This study consisted of 2 experimental visits organized at least 6 months apart.

### Preliminary questionnaires

At the beginning of visit 1, participants completed the short *Motion Sickness Susceptibility Questionnaire* (MSSQ-short) [30], the short version of the *Visually Induced Motion Sickness Susceptibility Questionnaire* (VIMSSQ-short) [31], and the *Immersive Tendencies Questionnaire* (ITQ) [32] in their French versions. These questionnaires were used in order to characterize past experiences with immersive technologies, and situations of transient discomfort.

### Ratings of visually evoked odor images

After completing the preliminary questionnaires at the start of visit 1, participants were asked to rate mentally simulated odor images. Seven pictures representing rose, violet, lavender, cucumber, strawberry, orange blossom and peppermint [20] were successively presented, and participants were instructed to concentrate on the odor evoked by each of these pictures. If they managed to generate a mental odor image, participants indicated the pleasantness and the intensity of this odor image on scales ranging from 1 to 4 (a high score indicating more pleasant/intense OI). Otherwise, they were instructed not to provide a rating. These data were collected in order to set individualized visual conditions during the VR immersion of visit 2 (see in the following section).

### Virtual immersions

During both visits, all participants performed a VR immersion in a simulated first-person view from a boat as described by Fantin et al. [7], equipped with an HTC Vive Pre (HTC, Taouyuan, Taïwan) head-mounted display. This virtual environment has been designed and previously used specifically to induce discomfort in experimental settings. Amplitude and frequencies of waves and simulated movements of the head in 3 dimensions make the environment visually unpredictable, facilitating the occurrence of a sufficient but tolerable level of cybersickness in more sensitive participants. The parameters used in this study are described in Fantin et al.'s original publication. Before each immersion, participants were asked to verbalize their current state of discomfort using the *Fast Motion Sickness Scale* (FMS) [33], ranging from 0 (absolutely no discomfort) to 20 (extreme discomfort). If a score strictly below 2/20 was verbalized, participants were instructed to complete the *Simulator Sickness Questionnaire* (SSQ) [6], which provided us with baseline information regarding their state before immersion. Then, the VR immersions could begin. Each immersion lasted a maximum of 14 minutes. Every minute, participants were asked to verbalize their state of discomfort using the FMS scale. If the verbalized score reached 16/20 or over, or if participants expressed the will to stop, the immersions were discontinued.

   Although the virtual environment used for both visits was identical, some visual information present in the visual field was manipulated. During visit 1 (Fig 1A), a black square frame

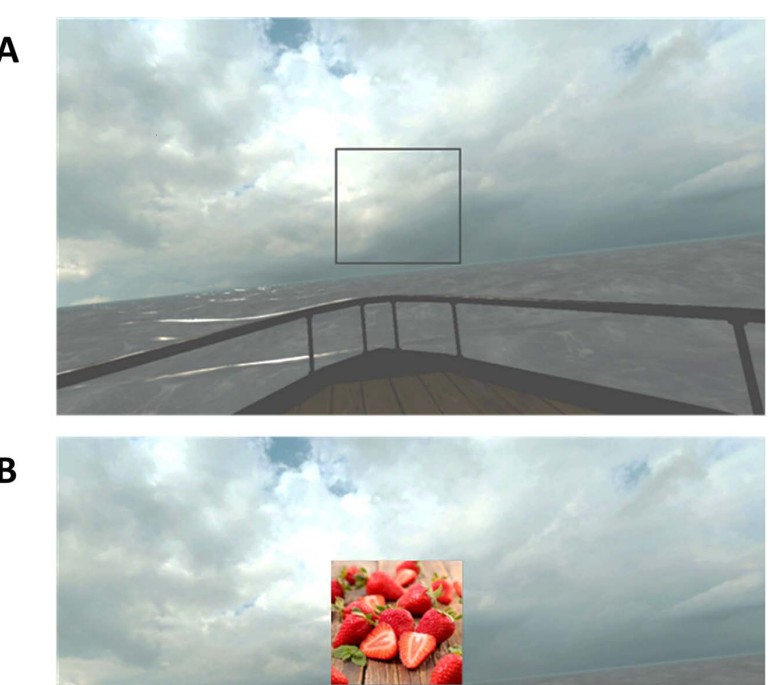

**Fig 1. Virtual environment used for the VR immersions.** A) visual setting of the control immersion (visit 1); B) associated to an OI task (visit 2, example of a picture representing strawberry).

(size = 17 degrees of the visual field) was placed at the center of the visual field and locked in order to stay centered despite any head movements. This frame was added to give participants static visual information on orientations, in order to control for the visual conditions of visit 2. Participants were informed they were allowed to place their gaze where they preferred but were also advised not to move their head in an unreasonable manner. During visit 2 (Fig 1 B), a picture among the 7 pictures rated during visit 1 was added to the center of the field of view (size = 17 degrees). The content of the picture was individually selected as having evoked the most pleasant and intense mental odor image during visit 1. The specific instruction given to participants for this visit was to concentrate on the odor evoked by the picture. It was specified that they were free not to look at the picture at all times, but that the picture was there to help them evoke the odor for as long as possible.

Directly after virtual removing the head-mounted display, participants completed the SSQ a second time, in order to assess the intensity of post-immersion cybersickness symptoms. The French version of the *Igroup Presence Questionnaire* (IPQ) [34] was also completed.

### Assessment of static field dependence

During both visits, static field dependence was assessed using the *Rod and Frame Test* (RFT). Participants were seated, equipped with the VR head-mounted display, and their head was held upright by a chinrest. In the VR headset, a 40 deg-tilted red rod was placed in an 18

deg-tilted gray frame. Using a videogame console controller, participants were instructed to rotate the rod until they judged it vertical (i.e., aligned with gravity). Eight trials were organized in 2 blocks of 4, corresponding to the 2 tilt directions of the frame (left or right). The direction of the initial tilt of the rod (left or right) was counterbalanced between blocks. There was no time limit, but participants were encouraged to answer spontaneously in a reasonable time frame. The RFT was performed before and after both virtual immersions. A representation of the time course of each visit is depicted in Fig 2.

### Data processing and statistical analysis

All data was stored in a pseudonymized database. For each visit, we considered and processed three indicators of user experience: SSQ scores, immersion duration, and sense of presence. Ratings for each completed SSQ were converted into a global score as described by Kennedy et al. [6]. For both visits, a $SSQ_{pre}$ and a $SSQ_{post}$ score were obtained from the pre and post-immersion total scores on the SSQ respectively. Then, $SSQ_{evol}$ scores were calculated for each visit by subtracting $SSQ_{pre}$ to $SSQ_{post}$ scores. A higher $SSQ_{evol}$ score indicates a broader effect of the virtual immersion on the self-declared intensity of cybersickness symptoms. Immersion duration corresponded to the number of minutes entirely performed in the virtual environment. Sense of presence scores corresponded to the IPQ global scores as described by Schubert et al. [34], higher scores indicating stronger sense of presence.

Scores obtained from the RFT were also processed for both visits. For each trial, we considered the angular difference between the objective vertical (i.e., direction of gravity) and the estimated vertical. Errors performed in the direction of tilt of the frame were noted positive, and errors in the opposite direction were noted as negative. Therefore, each participant obtained a $RFT_{pre}$ and $RFT_{post}$ score for each visit, calculated by averaging the 8 errors on pre

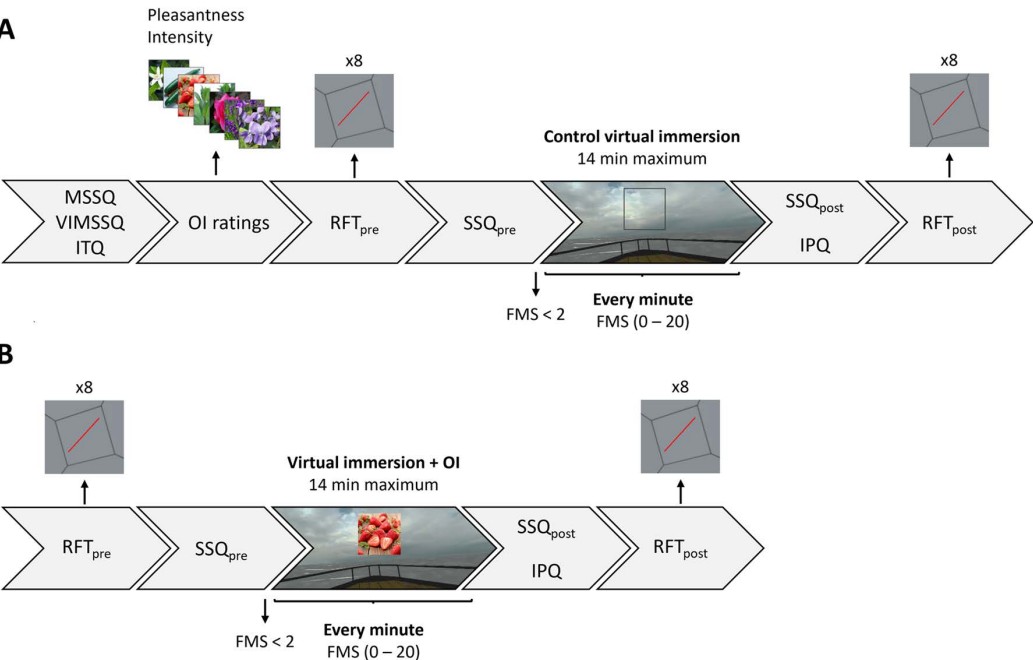

**Fig 2. Time course of both experimental visits.** A and B represent visits 1 and 2 respectively.

and post-immersion trials respectively. For both visits, a $RFT_{evol}$ score was obtained for each participant and each trial by subtracting $RFT_{pre}$ scores from $RFT_{post}$ scores.

Our statistical analysis first sought to investigate the effect of pleasant OI on cybersickness. To this end, we compared $SSQ_{evol}$ scores, immersion duration in minutes and IPQ scores between visits using statistical tests for paired samples.

Then, a linear mixed effects model was performed in order to determine which human or experimental factors contributed the most toward explaining $SSQ_{evol}$ scores. Because of our intrasubject experimental design, we entered participants as a random effect. Thus, the following model was constructed:

$$SSQ_{evol} = Visit + Gender + SSQ_{pre} + Visit : SSQ_{pre} + RFT_{pre} + Visit : RFT_{pre} + IPQ + RFT_{evol} + (1 \mid Participant)$$

We included $Visit{:}SSQ_{pre}$ and $Visit{:}RFT_{pre}$ interactions as fixed effects in order to account for any potential differences in initial states of our participants between visits, which were planned 6 months apart. Numerical data used in this model were reduced centered (converted to z-scores). Removal of one influential observation was performed after plotting standardized residuals against leverage. Collinearity, residual homoscedasticity, and residual linearity were verified. Although residuals presented heteroscedasticity, data were not further transformed given the little effect of this type of assumption violation on estimate bias in mixed effects models [35].

All statistical analyses were performed using Rstudio (version 2023.12.1) running on R (version 4.3.2). The mixed effects linear model was performed using the lmer function of the lme4 package [36]. Assumptions were verified using the performance package. Detailed results of the mixed effects model were obtained using the sjPlot package [37] (predictor estimates, confidence intervals and p-values as well as random effects) and performance package [38] (global performance of the model). The nature of statistical analyses for paired comparisons for the effect of session was defined after verifying normality of data using Shapiro-Wilk tests. The threshold for significance was set by default to 0.05. Quantitative data were therefore expressed either as means associated to standard deviation (SD), or medians (Med) associated to interquartile range (IQR). Qualitative data were expressed by their frequencies. All data used for the descriptive statistics or formal analyses are publicly available at https://doi.org/10.6084/m9.figshare.26927116.v1

## Results

Two participants did not perform the second visit since they declared having suffered from a smell-affecting illness (namely Covid) between visits, their data was therefore discarded. The descriptive data of the remaining 28 participants are available in Table 1.

Signed ranks Wilcoxon tests were performed to assess the effect of visit on $SSQ_{evol}$ scores, duration immersion and IPQ scores (Fig 3). Our results show that for $SSQ_{evol}$ scores, participants presenting were significantly lower scores after the visit with OI (Med = 9.35; IQR = 33.7) compared to the control visit (Med = 35.5; IQR = 74.8; $p$ = 0.038; effect size = 0.37). Moreover, immersion durations were significantly higher during OI (Med = 14; IQR = 0) than during the control visit (Med = 14; IQR = 7.25; $p$ = 0.002; effect size = 0.65). Finally, our analysis did not reveal any significant effect of visit on IPQ scores.

The mixed effects model used to explain $SSQ_{evol}$ scores revealed that only the effect of visit was statistically significant ($p$ = 0.01). The Akaike Information Criterion (AIC) of the model was 504.42. Details on the model are provided in Table 2

**Table 1. Descriptive data of the population before the control VR immersion.**

| Characteristics | N = 28 |
|---|---|
| Gender | |
| Male | 15 [53.6%] |
| Female | 13 [46.4%] |
| Age | 23 (4) \| 19–33 |
| fVOIQ | 1.94 (0.35) \| 1.31–2.44 |
| MSSQ-short | 15 (10) \| 0–34 |
| VIMSSQ-short | 6 (3) \| 0–12 |
| ITQ | 93 (13) \| 68–114 |
| $RFT_{pre}$ (Visit 1) | 4.80 (3.12) \| 0.90–13.95 |
| N [%]; Mean (SD) \| Min–Max | |

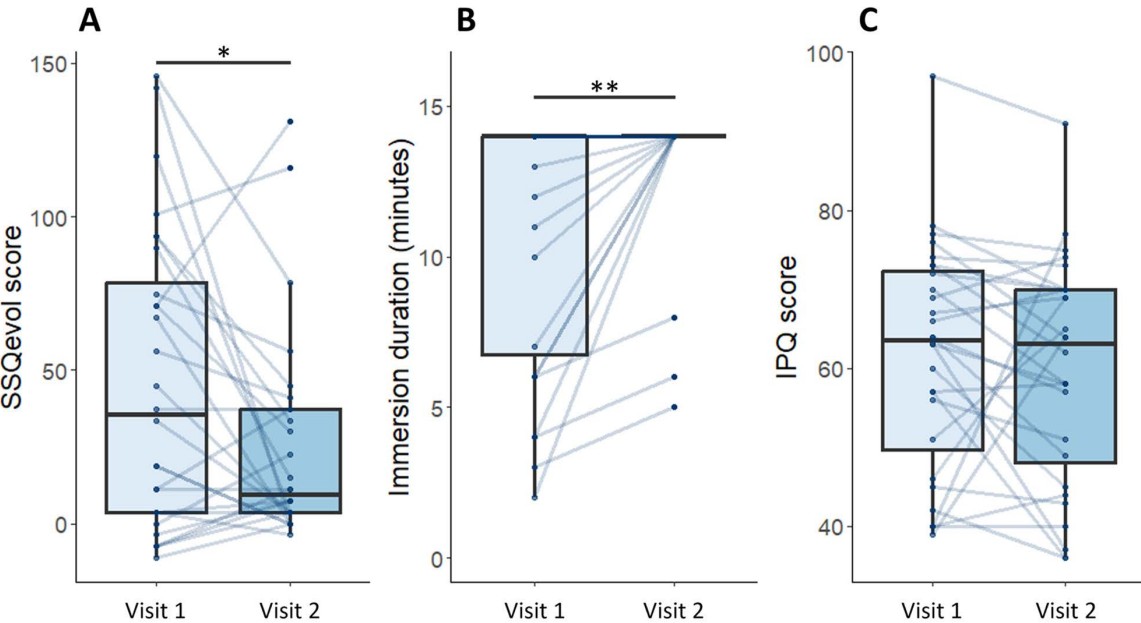

**Fig 3. Boxplot representations of user experience variables as a function of visit.** Pairwise comparisons were performed on A) $SSQ_{evol}$ scores, B) immersion durations and C) IPQ scores (*: $p < 0.05$; **: $p < 0.01$). Medians and quartiles are represented, points and lines indicate individual values.

## Discussion

This study proposed an innovative environmental-based solution for alleviating cybersickness in VR, using pleasant visuo-olfactory evocation. Our results corroborate those observed in the case of pleasant odor perception [15], as we show a decrease $SSQ_{evol}$ scores, associated to an increased immersion duration. In other words, not only did participants experience less intense cybersickness symptoms when generating pleasant odor images, but they also tolerated the virtual environment for a longer period. The lower $SSQ_{evol}$ scores observed during this second visit therefore cannot be attributed to premature exit from the virtual stimulation. Moreover, although the observed effects are subject to residual interindividual variability certainly due to other factors which were not measured, our mixed effects model showed that the effect of session was the most explanatory of the $SSQ_{evol}$ scores, and our only statistically

**Table 2. Detailed values of predictors and random effects for the linear mixed effects model explaining SSQ$_{evol}$ scores.**

| Predictors | Beta | 95% confidence interval | p |
|---|---|---|---|
| Constant | 44.26 | [21.22; 67.30] | <0.001 |
| Visit [2]$^a$ | −20.88 | [−38.24; −3.52] | 0.02 |
| Gender [Men] | −2.64 | [−32.21; 26.93] | 0.86 |
| SSQ$_{pre}$ | −2.09 | [−15.08; 10.90] | 0.75 |
| Visit [2] * SSQ$_{pre}$ | −4.96 | [−23.87; 13.96] | 0.60 |
| RFT$_{pre}$ | −17.81 | [−36.10 ;0.48] | 0.06 |
| Visit [2] * RFT$_{pre}$ | 9.86 | [−7.35 ;27.08] | 0.25 |
| IPQ | 7.46 | [−4.58; 19.49] | 0.22 |
| RFT$_{evol}$ | −7.34 | [−17.32; 2.63] | 0.15 |
| Random effects | | | |
| σ² | 746.51 | | |
| τ$_{00}$ Participant | 924.67 | | |
| ICC | 0.55 | | |
| Marginal R² | 0.14 | | |
| Conditional R² | 0.62 | | |

$^a$Visit [2] = Visit including OI during virtual immersion

significant fixed effect. Thus, pleasant OI seems to be a method allowing for alleviation of cybersickness in more sensitive users, while not negatively affecting user experience in those less prone to discomfort.

Given the numerous similarities reported between odor perception and OI, it is possible that the mechanisms explaining the alleviation of cybersickness in both cases are comparable, that is to say mediated by emotional processes [16]. Indeed, the reported involvement of brain regions during OI such as the insula and the DLPFC [25–27] do suggest that many processes related to emotional regulation take place when this task is successfully performed. Therefore, we can suppose that pleasant OI elicited specific emotional states leading to a decrease in cybersickness, although the mechanisms underpinning this effect are still not certain. The interpretation proposed in the case of odor diffusion [15,16] is anchored in the modal model of emotional regulation [39] consisting in the succession of situation, attention, appraisal, and response. According to this model, appraisal is understood as the meaning given to the situation by an individual, which then manifests as "*a coordinated yet malleable multisystem response*" in the context of the situation [40]. The decrease in cybersickness would therefore be one tangible consequence of this multisystem response taking place at a physiological level, although this remains to be demonstrated using more direct measures. For instance, it would be relevant investigate biological indicators such as those obtained from heart rate variability, in the light of the well-known interactions between the brain and heart occurring in situations of emotional regulation [41]. One other arising question is whether such emotional states can be obtained from imagery in other sensory modalities. It has indeed been thoroughly discussed that OI is particularly difficult compared to sensory imagery in other modalities [19,42,43]. Therefore, while our results are very encouraging and open new perspectives for the alleviation of cybersickness based on sensory imagery, one potential limitation of this approach could be related to the necessary cognitive load, in particular when performing a task simultaneously. However, if multisystem responses as described in the modal model can be obtained from emotional states regardless of how they are inducted, perhaps pleasant using visual or auditory imagery could prove more efficient. Because

these are known to be easier to perform and concern a greater proportion of the population, such an approach could help reduce cognitive load and improve the applicability of such a strategy in ecological settings. Further studies should therefore seek to determine whether the outcomes of our study are specific to olfaction or are transposable to other sensory modalities.

It is nevertheless possible that other factors may have contributed to the effects observed in our work. For example, it is known that adding a visual anchorage in virtual environments can contribute to decreasing symptoms of cybersickness by reducing the amount of visual flux users are exposed to. Perhaps in our study, the pictures used during the second VR immersion acted as an anchorage. In the control visit, we attempted to control for this by adding a still frame in the center of the visual field. While this method does provide static visual information about verticality and horizontality, we cannot assert that it has identical effects to pictures since visual flux was present inside the frame contrarily to pictures which were opaque. However, IPQ scores did not indicate any significant increase or decrease in sense of presence between visits, suggesting no effect of placing a picture in the visual field. Secondly, one could wonder whether the effect of OI is simply driven by reallocating participants' attention towards the OI task rather than the virtual world, regardless of the user's subsequent emotional state. In this case, pleasant and unpleasant OI would have similar effects on cybersickness. In order to investigate these two alternative interpretations (visual flux and attention reallocation), we are currently carrying out a complementary study using different conditions of visual anchorage and OI hedonicity. Preliminary results on 19 participants (not shown here) suggest a decrease in cybersickness compared to a control condition, only when OI is rated as pleasant. Although it remains to be confirmed, it seems that our preliminary results are therefore in favor of the hypothesis of emotional regulation, which remains to be confirmed by measures of autonomic activity. If so, these first findings may open the door to novel approaches for the mitigation of transient states of discomfort using induction of emotional states, not only in the context of immersive technologies but perhaps also in other settings such as motion sickness in means of transportation.

## Conclusion

The present study aimed at determining the effect of pleasant OI on VR-induced cybersickness. Our main results show a decrease in the intensity of cybersickness symptoms when participants performed OI, associated with a longer tolerance of the virtual environment. These encouraging findings suggest the potential relevance of self-induced emotional states to improve well-being during virtual immersions. One remaining issue which should motivate future work resides in the cognitive load necessary for OI, possibly negatively affecting users' performance at primary tasks that require active involvement (e.g., navigation or physical interactions). If OI reveals itself too problematic, our results still open the door toward the use of other forms of sensory imagery or less costly cognitive mechanisms to alleviate transient states of discomfort, whether they are in virtual environments or not.

## Author contributions

**Conceptualization:** Luca Fantin, Gabriela Hossu, Hadrien Ceyte.

**Data curation:** Luca Fantin, Gabriela Hossu, Hadrien Ceyte.

**Formal analysis:** Luca Fantin, Gabriela Hossu, Hadrien Ceyte.

**Funding acquisition:** Gabriela Hossu.

**Investigation:** Luca Fantin.

**Methodology:** Luca Fantin, Gabriela Hossu, Guillaume Drouot, Hadrien Ceyte.

**Project administration:** Luca Fantin, Gabriela Hossu, Cécile Rumeau, Guillaume Drouot, Hadrien Ceyte.

**Resources:** Cécile Rumeau, Guillaume Drouot.

**Software:** Luca Fantin, Gabriela Hossu, Hadrien Ceyte.

**Supervision:** Luca Fantin, Gabriela Hossu, Hadrien Ceyte.

**Validation:** Gabriela Hossu, Cécile Rumeau, Hadrien Ceyte.

**Visualization:** Luca Fantin, Gabriela Hossu.

**Writing – original draft:** Luca Fantin, Gabriela Hossu, Hadrien Ceyte.

**Writing – review & editing:** Luca Fantin, Gabriela Hossu, Cécile Rumeau, Guillaume Drouot, Hadrien Ceyte.

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
