## [Decision Letter · Decision Letter 0]

21 Oct 2024

PONE-D-24-38671Is it all in your head? Reducing virtual reality induced cybersickness by pleasant odor imageryPLOS ONE

Dear Dr. Ceyte,

Thank you for submitting your manuscript to PLOS ONE. After careful consideration, we feel that it has merit but does not fully meet PLOS ONE’s publication criteria as it currently stands. Therefore, we invite you to submit a revised version of the manuscript that addresses the points raised during the review process.

**ACADEMIC EDITOR: Justify the research question**

We look forward to receiving your revised manuscript.

Kind regards,

Jeyasakthy Saniasiaya, MD, MMed ORLHNS, FEBORLHNS

Academic Editor

PLOS ONE

Journal Requirements: When submitting your revision, we need you to address these additional requirements. 1. Please ensure that your manuscript meets PLOS ONE's style requirements, including those for file naming. The PLOS ONE style templates can be found at https://journals.plos.org/plosone/s/file?id=wjVg/PLOSOne_formatting_sample_main_body.pdf and https://journals.plos.org/plosone/s/file?id=ba62/PLOSOne_formatting_sample_title_authors_affiliations.pdf 2. Thank you for stating the following financial disclosure: "This work is co-funded by the French State-Region contract CPER 2015-2020 (Contrat de Plan Etat Région– IT2MP Innovations Technologiques, Modélisation et Médecine Personnalisée), by the European Union through the European Regional Development Fund “FEDER-FSE Lorraine et Massif des Vosges 2014-2020” and by the CHRU of Nancy, France."  Please state what role the funders took in the study.  If the funders had no role, please state: ""The funders had no role in study design, data collection and analysis, decision to publish, or preparation of the manuscript."" If this statement is not correct you must amend it as needed. Please include this amended Role of Funder statement in your cover letter; we will change the online submission form on your behalf. 3. Thank you for stating the following in the Acknowledgments Section of your manuscript: "This work is co-funded by the French State-Region contract CPER 2015-2020 (Contrat de Plan Etat Région– IT2MP Innovations Technologiques, Modélisation et Médecine Personnalisée), by the European Union through the European Regional Development Fund “FEDER-FSE Lorraine et Massif des Vosges 2014-2020” and by the CHRU of Nancy, France." We note that you have provided funding information that is not currently declared in your Funding Statement. However, funding information should not appear in the Acknowledgments section or other areas of your manuscript. We will only publish funding information present in the Funding Statement section of the online submission form. Please remove any funding-related text from the manuscript and let us know how you would like to update your Funding Statement. Currently, your Funding Statement reads as follows: "This work is co-funded by the French State-Region contract CPER 2015-2020 (Contrat de Plan Etat Région– IT2MP Innovations Technologiques, Modélisation et Médecine Personnalisée), by the European Union through the European Regional Development Fund “FEDER-FSE Lorraine et Massif des Vosges 2014-2020” and by the CHRU of Nancy, France." Please include your amended statements within your cover letter; we will change the online submission form on your behalf. 4. Please provide a complete Data Availability Statement in the submission form, ensuring you include all necessary access information or a reason for why you are unable to make your data freely accessible. If your research concerns only data provided within your submission, please write "All data are in the manuscript and/or supporting information files" as your Data Availability Statement. 

**Additional Editor Comments:**

Justify how this study adds to the current knowledge

Reviewers' comments:

Reviewer's Responses to Questions

**Comments to the Author**

1. Is the manuscript technically sound, and do the data support the conclusions?

Reviewer #1: No

Reviewer #2: Yes

2. Has the statistical analysis been performed appropriately and rigorously? 

Reviewer #1: Yes

Reviewer #2: Yes

3. Have the authors made all data underlying the findings in their manuscript fully available?

Reviewer #1: Yes

Reviewer #2: Yes

4. Is the manuscript presented in an intelligible fashion and written in standard English?

Reviewer #1: Yes

Reviewer #2: Yes

5. Review Comments to the Author

Reviewer #1: This is an interesting study investigating the effects of odor imagery on cybersickness. Using a within-subject design, the authors examined whether individuals (N = 28) demonstrate changes in cybersickness across two timepoints (with 6 months apart), having watched images of pleasant fruit (different for each subject) at T2. The results indicate that cybersickness was significantly lower during the 2nd visit, where odor imagery was displayed during the VR task (as opposed to a black square during the VR task for the 1st visit). The study is concise, largely well-written, and focuses on an interesting topic. There are some caveats, however, detailed below.

• The authors should provide a clear justification for the use of within- rather than a between-subjects design. The reason for this is that it is not possible to deduce that the reduction in cybersickness was due to odor imagery, as there should have been a comparison or a control group not receiving the OI.

• In addition, a priori power analysis should have been conducted and reported, but if this was not done, then sensitivity analysis would be useful considering the small sample. Having run a quick power analysis using similar parameters, the current sample size appears to be too small (my results yielded a minimum sample of 36). The authors should expand on this.

• The authors should justify the use of their VR immersion scenario. It sounds like the scenario was designed to induce cybersickness (in addition, the authors should provide more details about this, rather than just a reference), and that participants were asked to sit still and not tilt their head too much Nevertheless, cybersickness often appears in more innocuous scenarios (e.g., walking around, peering down, etc.). How does the study scenario compare to cybersickness induced by more typical usage (e.g., during video game play)? This should be clarified.

• Methods: there are details lacking about psychometric properties of the scales used, the type of standardization method used for numerical data, and how outlier removal was conducted (and why it was removed rather than using other methods, considering the small sample size).

• A major limitation regarding the study conclusions is the effect of familiarity using the VR the 2nd time around. Although the participants were asked about their motion sickness susceptibility and prior engagement with technology, having experienced the VR scenario once does create familiarity with the environment the next time around, which could impact on cybersickness (see e.g., Petri et al., 2020).

• On a minor note, I suggest the paper is proof-read by an English speaker, as there are minor grammatical errors throughout (e.g., Introduction, 1st paragraph, the phrasing “Provided the place of VR…” is awkward wording).

References

Petri, K., Feuerstein, K., Folster, S., Bariszlovich, F., & Witte, K. (2020). Effects of Age, Gender, Familiarity with the Content, and Exposure Time on Cybersickness in Immersive Head-mounted Display Based Virtual Reality. American Journal of Biomedical Sciences, 12(2),107-121.

Reviewer #2: Thank you for the opportunity to read this article. This paper touches on an important topic and is well-written. Some minor comments for the author's consideration:

1) Is there a sample size calculation?

2) Would this lead to another distraction if participants were asked to immediately describe their state of discomfort?

3) How to judge whether the immersion will stop when the score reaches 16/20?

4) It is recommended to add a limitation and conclusion section.

6. PLOS authors have the option to publish the peer review history of their article (what does this mean? ). If published, this will include your full peer review and any attached files.

**Do you want your identity to be public for this peer review?** For information about this choice, including consent withdrawal, please see our Privacy Policy .

Reviewer #1: No

Reviewer #2: No

---

## [Author Response · Author response to Decision Letter 1]

2 Dec 2024

REVIEWERS' COMMENTS

Reviewer #1:

This is an interesting study investigating the effects of odor imagery on cybersickness. Using a within-subject design, the authors examined whether individuals (N = 28) demonstrate changes in cybersickness across two timepoints (with 6 months apart), having watched images of pleasant fruit (different for each subject) at T2. The results indicate that cybersickness was significantly lower during the 2nd visit, where odor imagery was displayed during the VR task (as opposed to a black square during the VR task for the 1st visit). The study is concise, largely well-written, and focuses on an interesting topic. There are some caveats, however, detailed below.

The authors should provide a clear justification for the use of within- rather than a between-subjects design. The reason for this is that it is not possible to deduce that the reduction in cybersickness was due to odor imagery, as there should have been a comparison or a control group not receiving the OI.

Author’s response:

We thank Reviewer #1 for this insight. The choice of a within-subjects design was motivated by our main question, which was the improvement of well-being in VR, by using individually-selected visual cues.

We understand the Reviewer’s concern on the absence of a control group. However, it is well known that susceptibility to cybersickness is extremely variable between individuals, and depends on many factors (some of which are difficult if not impossible to control). For this reason, pairing a control group based on a prediction of their sensitivity to our virtual environment would have been extremely complex. The risk would be to obtain controls which respond differently to our virtual environment, questioning the relevance of a between-groups comparison. Consequently (and for ethical considerations which are addressed in the Reviewer’s next point, see below) we focused on a within-subject design.

However, we do agree with the Reviewer about the fact that other factors may have contributed to the decrease in cybersickness which we observed in our study. These factors (i.e., the presence of a visual anchorage, and other mechanisms of attentional reallocation) are addressed in our discussion section. Our future work, which already wields interesting preliminary results presented in our discussion, will focus on these questions and provide more insight regarding the role of odor imagery specifically, relatively to other potential confounding factors.

Reviewer’s comment:

In addition, a priori power analysis should have been conducted and reported, but if this was not done, then sensitivity analysis would be useful considering the small sample. Having run a quick power analysis using similar parameters, the current sample size appears to be too small (my results yielded a minimum sample of 36). The authors should expand on this.

Author’s response:

The sample size for this study was determined based on the nature of our study rather than a power analysis. The clinical trial which the present study was part of (CANOE NCT05308433), includes an fMRI session in which all participants took part. Because this project was exploratory, ethical requirements were to determine the sample based on the main goal of the clinical trial, which was in fMRI. This limited us to a sample size of 30 participants.

We understand the Reviewer’s concerns regarding the sample size, which we share. This is why our statistical analyses focused on a within-subjects design, in an attempt to maximize our statistical power. In addition, effect sizes are specified in the manuscript.

Reviewer’s comment:

The authors should justify the use of their VR immersion scenario. It sounds like the scenario was designed to induce cybersickness (in addition, the authors should provide more details about this, rather than just a reference), and that participants were asked to sit still and not tilt their head too much. Nevertheless, cybersickness often appears in more innocuous scenarios (e.g., walking around, peering down, etc.). How does the study scenario compare to cybersickness induced by more typical usage (e.g., during video game play)? This should be clarified.

Author’s response:

We chose to use the virtual environment described in Fantin et al. (2023), since it has been validated in experimental settings and offers interesting features to control cybersickness to a certain extent. The following parameters are described in the original publication, and were chosen following several pre-tests:

- Waves were set at 0.7m of amplitude at two frequencies of 0.3Hz and 0.5Hz.

- Simulated head movements were set at:

o Pitch and yaw: 4deg amplitude at a frequency of 0.1Hz.

o Roll: 7deg amplitude, at 0.4Hz.

We understand that readers may want more information in our article, which is why we have added some justification to our choice of stimulation (page 6, lines 127-132) : “This virtual environment has been designed and previously used (…) original publication”

For the precise settings of frequency and amplitude of waves and simulated head movements, we encourage readers to refer to the original publication (Fantin et al., 2023).

Moreover, we are fairly certain that this scenario can compare to cybersickness in other types of virtual environment. Cybersickness is usually seen as the result of a cognitive-sensory conflict, which was the case in our study even though participants were stationary and passive. The use of this “passive” scenario allowed for participants to experiment the virtual environment similarly and concentrate on odor imagery with no confound related to their expertise in gaming. In our discussion section, we do however discuss the possible difficulties that users could encounter if they were asked to perform odor imagery while simultaneously playing a game in a virtual environment. This limit related to cognitive load has been acknowledged and should be addressed in future studies.

Reviewer’s comment:

Methods: there are details lacking about psychometric properties of the scales used, the type of standardization method used for numerical data, and how outlier removal was conducted (and why it was removed rather than using other methods, considering the small sample size).

Author’s response:

The scales for descriptive statistics (MSSQ, VIMSSQ, ITQ) and for our formal analyses (SSQ, IPQ) are validated tools. Our work referenced each of the original publications which describe their properties and how scores are calculated.

The following information has been added accordingly with the Reviewer’s suggestions:

- Numerical data were centered reduced (page 10, line 209)

- Removal of one influential observation was performed after plotting standardized residuals against leverage (page 10, lines 210-211)

Outlier removal was performed rather than other methods since it offered the best compromise between the respect of statistical assumptions, and interpretability in the context of our questioning. This is especially true since it only resulted in the exclusion of 1 participant, which we considered acceptable for this exploratory analysis.

We thank Reviewer #1 for pointing out these omissions and we hope that this answers their concerns.

Reviewer’s comment:

A major limitation regarding the study conclusions is the effect of familiarity using the VR the 2nd time around. Although the participants were asked about their motion sickness susceptibility and prior engagement with technology, having experienced the VR scenario once does create familiarity with the environment the next time around, which could impact on cybersickness (see e.g., Petri et al., 2020).

Author’s response:

We thank Reviewer #1 for raising this point. If our understanding of Petri et al.’s study is correct, “familiarity” in this case familiarity with content of the depicted task (in this case experts in karate vs non-experts), rather than being familiar with a specific virtual environment.

We are aware that there have been some studies on the potential impact of repeated exposure to VR, namely cybersickness abatement by repeated exposure (CARE). However, these methodologies rely on multiple VR exposures with little rest time (sessions are usually organized on successive days) and use complementary manipulation of visual stimulation, such as peripheral FoV reduction. Outside of these specific studies, most reports of adaptation to VR take place over a much longer time scale.

In our study, participants only performed two experimental sessions which were purposely set 6 months apart to limit any potential effect of the first session on the second. Moreover, adding to odor imagery task during the second session changed visual and cognitive constraints applied on participants, further reducing any effect of familiarity to the task and the environment. For these reasons, we believe that the mitigation of cybersickness we observe is not due to any form of familiarity.

Reviewer’s comment:

On a minor note, I suggest the paper is proof-read by an English speaker, as there are minor grammatical errors throughout (e.g., Introduction, 1st paragraph, the phrasing “Provided the place of VR…” is awkward wording).

The article has been proof-read and minor corrections have been applied throughout the manuscript. We thank the Reviewer for their insight.

Reviewer #2:

Thank you for the opportunity to read this article. This paper touches on an important topic and is well-written. Some minor comments for the author's consideration:

Reviewer’s comment:

Is there a sample size calculation?

Author’s response:

As stated in the responses to Reviewer #1, sample size for this study was not determined based on a power analysis. The clinical trial which the present study was part of (CANOE NCT05308433), includes an fMRI session in which all participants took part. Because this project was exploratory, ethical requirements were to determine the sample based on the main goal of the clinical trial, which was in fMRI. This limited us to a sample size of 30 participants.

Reviewer’s comment:

Would this lead to another distraction if participants were asked to immediately describe their state of discomfort?

Author’s response:

If we understand correctly, Reviewer #2 suggests that we ask participants to describe their discomfort qualitatively rather than using a number during the virtual immersions.

While it is an interesting suggestion, it does in our opinion have some methodological issues. Mainly, it would be difficult to control the amount of detail with which each participant would describe this discomfort. On the contrary, verbalizing numbers allows for more homogenized speech time. It is also possible that verbally describing one’s state of discomfort may increase one’s focus toward these symptoms, and as a consequence increase them.

Therefore, we decided to use the Fast Motion Sickness Scale in line with previous works, using it only as a control in order to discontinue virtual immersions when needed.

Reviewer’s comment:

How to judge whether the immersion will stop when the score reaches 16/20?

Author’s response:

Most previous works on cybersickness or other forms of transient discomfort have used scales in order to regularly control participant’s state. It is also an important ethical consideration that participants do not experience an unnecessary amount of discomfort.

Generally, these studies discontinue stimulation when participants reach a sufficient level of discomfort, corresponding to values around 75% of the maximum : 16/20 for the FMS (Fantin et al., 2023; Keshavarz B & Hecht H, 2011), or 4/6 for scales with lower resolution such as the Sickness Rating scale (Cian et al., 2011).

The threshold used in our study was also intended to be close to that of the original authors of the FMS : “… FMS scores were verbally requested and noted by the experimenter once a minute. The very last score was reported at the moment of video fade-out or just before the experiment was prematurely aborted. The latter could occur at the participant’s request or if the experimenter considered the MS score to be indicative of impending frank sickness. When a score of 15 had been reached, the experimenter deliberately asked the participant if he or she wanted to abort the experiment.”(Keshavarz & Hecht, 2011)

In any case, the virtual immersions were discontinued whenever participants expressed the will to stop, as specified in our methods section.

References :

Cian, C., Ohlmann, T., Ceyte, H., Gresty, M. A., & Golding, J. F. (2011). Off vertical axis rotation motion sickness and field dependence. Aviation, Space, and Environmental Medicine, 82(10), 959-963.

Fantin, L., Ceyte, G., Maïni, E., Hossu, G., & Ceyte, H. (2023). Do individual constraints induce flexibility of visual field dependence following a virtual immersion? Effects of perceptive style and cybersickness. Virtual Reality, 27(2), 917-928.

Keshavarz, B., & Hecht, H. (2011). Validating an efficient method to quantify motion sickness. Human factors, 53(4), 415-426.

Reviewer’s comment:

It is recommended to add a limitation and conclusion section.

Author’s response:

We thank Reviewer #2 for this observation. We have added a conclusion section containing the key messages of our study.

The limits of our study do appear throughout the discussion section rather than in a separate section, which was an intentional choice on our part. We believe it is relevant to use them to nuance the main results that we present and discuss. We hope that this choice will still satisfy the Reviewer’s demands.

---

## [Decision Letter · Decision Letter 1]

6 Feb 2025

Is it all in your head? Reducing virtual reality induced cybersickness by pleasant odor imagery

PONE-D-24-38671R1

Dear Dr. Ceyte,

We’re pleased to inform you that your manuscript has been judged scientifically suitable for publication and will be formally accepted for publication once it meets all outstanding technical requirements.

Kind regards,

Jeyasakthy Saniasiaya, MD, MMed ORLHNS, FEBORLHNS

Academic Editor

PLOS ONE

Additional Editor Comments (optional):

Revisions has been adequately addressed

Reviewers' comments:

Reviewer's Responses to Questions

**Comments to the Author**

1. If the authors have adequately addressed your comments raised in a previous round of review and you feel that this manuscript is now acceptable for publication, you may indicate that here to bypass the “Comments to the Author” section, enter your conflict of interest statement in the “Confidential to Editor” section, and submit your "Accept" recommendation.

Reviewer #2: All comments have been addressed

2. Is the manuscript technically sound, and do the data support the conclusions?

Reviewer #2: Yes

3. Has the statistical analysis been performed appropriately and rigorously? 

Reviewer #2: Yes

4. Have the authors made all data underlying the findings in their manuscript fully available?

Reviewer #2: Yes

5. Is the manuscript presented in an intelligible fashion and written in standard English?

Reviewer #2: Yes

6. Review Comments to the Author

Reviewer #2: Thank you for your revision! The author has responded to all my comments.

I have no other comments.

7. PLOS authors have the option to publish the peer review history of their article (what does this mean? ). If published, this will include your full peer review and any attached files.

**Do you want your identity to be public for this peer review?** For information about this choice, including consent withdrawal, please see our Privacy Policy .

Reviewer #2: **Yes: ** Cho Lee Wong

---

## [Editor Report · Acceptance letter]

PONE-D-24-38671R1

PLOS ONE

Dear Dr. Ceyte,

I'm pleased to inform you that your manuscript has been deemed suitable for publication in PLOS ONE. Congratulations! Your manuscript is now being handed over to our production team.

Kind regards,

on behalf of

Dr. Jeyasakthy Saniasiaya

Academic Editor

PLOS ONE